# Diff-BBO: Diffusion-Based Inverse Modeling for Black-Box Optimization

**Dongxia Wu**
University of California San Diego
La Jolla, CA
dowu@ucsd.edu

**Nikki Lijing Kuang**
University of California San Diego
La Jolla, CA
l1kuang@ucsd.edu

**Ruijia Niu**
University of California San Diego
La Jolla, CA
rniu@ucsd.edu

**Yi-An Ma**
University of California San Diego
La Jolla, CA
yianma@ucsd.edu

**Rose Yu**
University of California San Diego
La Jolla, CA
roseyu@ucsd.edu

## Abstract

Black-box optimization (BBO) aims to optimize an objective function by iteratively querying a black-box oracle in a sample-efficient way. While prior studies focus on forward approaches to learn surrogates for the unknown objective function, they struggle with steering clear of out-of-distribution and invalid inputs. Recently, inverse modeling approaches that map objective space to the design space with conditional diffusion models have demonstrated impressive capability in learning the data manifold. They have shown promising performance in offline BBO tasks. However, these approaches require a pre-collected dataset. How to design the acquisition function for inverse modeling to *actively* query new data remains an open question. In this work, we propose *diffusion-based inverse modeling for black-box optimization* (Diff-BBO), an *inverse* approach leveraging diffusion models for online BBO problem. Instead of proposing candidates in the design space, Diff-BBO employs a novel acquisition function *Uncertainty-aware Exploration* (UaE) to propose objective function values. Subsequently, we employ a conditional diffusion model to generate samples based on these proposed values within the design space. We demonstrate that using UaE results in optimal optimization outcomes, supported by both theoretical and empirical evidence.

## 1 Introduction

Practical problems in science and engineering often involve optimizing a black-box objective function that is expensive to evaluate, such as robotics (Tesch et al., 2013) and molecular design (Sanchez-Lengeling and Aspuru-Guzik, 2018). How to achieve a near-optimal solution while minimizing function evaluations is thus a major challenge in black-box optimization (BBO). Prior works in BBO have largely focused on the online setting where a model can iteratively query the function during training (Turner et al., 2021; Zhang et al., 2021; Hebbal et al., 2019; Mockus, 1974). Most existing algorithms belong to forward methods, including Bayesian optimization (BO) (Kushner, 1964; Mockus, 1974; Wu et al., 2023; Frazier, 2018), bandit algorithms (Agrawal and Goyal, 2012;

Workshop on Bayesian Decision-making and Uncertainty, 38th Conference on Neural Information Processing Systems (NeurIPS 2024).

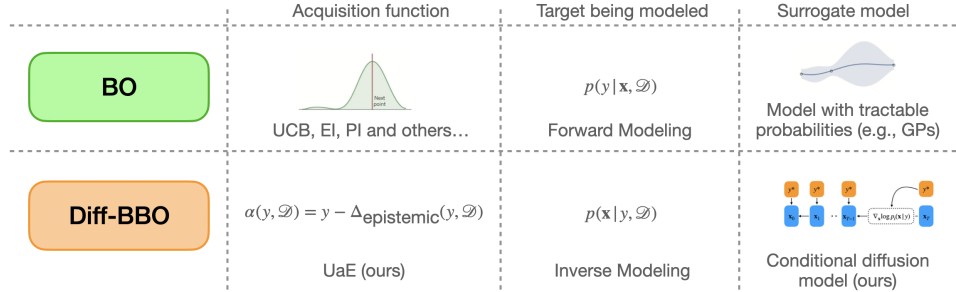

Figure 1: Forward modeling vs inverse modeling for black-box optimization. (*Top*) Forward modeling approach using certain surrogate models (e.g., GPs) for forward modeling and acquisition functions (e.g., UCB, PI, and EI) to select $\mathbf{x}$. (*Bottom*) Our inverse modeling approach using generative model (e.g., diffusion model) for inverse modeling and acquisition function (e.g., UaE) to select $y$.

Karbasi et al., 2023), and conditional sampling approaches (Brookes et al., 2019; Gruver et al., 2024; Stanton et al., 2022). They build a surrogate model to approximate the black-box function and optimize sequentially.

However, these approaches may face difficulties in scenarios where valid inputs represent a small subspace, such as valid protein sequences or molecular structures. Such optimization problems become exceptionally challenging, as the optimizer must navigate and avoid out-of-distribution and invalid inputs (Kumar and Levine, 2020). Recently, a novel set of methods, termed *inverse approaches*, have been proposed to address this issue. These methods (Kumar and Levine, 2020; Krishnamoorthy et al., 2023; Kim et al., 2023; Fu and Levine, 2021) break the traditional paradigm by learning an inverse mapping from objective space back to the input (design) space. Leveraging the state-of-the-art generative models, such as diffusion models (Sohl-Dickstein et al., 2015; Song et al., 2020), these approaches effectively capture data distributions in high-dimensional input space and facilitate optimization within the data manifold (Kong et al., 2024; Li et al., 2024). They achieve high performance in offline optimization settings (Kumar and Levine, 2020; Lu et al., 2023; Wang et al., 2018), assuming access to a fixed pre-collected dataset.

Despite these advancements, the online setting of inverse modeling, particularly how to capture the uncertainty of the inverse model and design an acquisition function for data-efficient querying, remains an open question. In this paper, we propose Diff-BBO, an inverse approach for online black-box optimization. Our approach consists of a novel acquisition function design through the *uncertainty quantification* (UQ) of conditional diffusion model, which proposes the desired objective function values to strategically sample the design space and query the oracle function efficiently. We summarize our main contributions as follows:

- We present Diff-BBO, an inverse modeling approach for efficient online black-box optimization (BBO) leveraging uncertainty of conditional diffusion models.

- We design a novel acquisition function for BBO based on the uncertainty of conditional diffusion models. Theoretically, we prove that the balance between targeting higher objective values and minimizing epistemic uncertainty lead to optimal optimization outcomes.

- We demonstrate that Diff-BBO achieves state-of-the-art performance with superior sample efficiency on Design-Bench and molecular discovery task in the online BBO setting.

## 2 Methodology

Let $f : \mathcal{X} \to \mathbb{R}$ denote the unknown ground-truth black-box function that evaluates the quality of any data point $\boldsymbol{x}$, with $\mathcal{X} \subseteq \mathbb{R}^d$. Our goal is to find the optimal point $\boldsymbol{x}^*$ that maximizes $f$, $\boldsymbol{x}^* \in \operatorname{argmax}_{\boldsymbol{x} \in \mathcal{X}} f(\boldsymbol{x})$. In the batch online BBO setting, we iteratively query $f$ with batch size $N$ and a fixed query iteration $K$ and update the model based on observed outputs. At each iteration, the acquisition function guides the data selection of new query points by balancing exploration and exploitation. In Diff-BBO, we model the conditional distribution of $p(\boldsymbol{x}|y, \mathcal{D})$ with training data $\mathcal{D}$. The function value $y$ to condition on is proposed by an acquisition function, which quantifies the

quality of the generated $\boldsymbol{x}$. The objective of the above optimization becomes:

$$\max_{y_k \in \mathbb{R}} \sum_{k=1}^{K} f(\boldsymbol{x}_k), \ \ \boldsymbol{x}_k \sim p_\theta(\cdot \mid y_k, \mathcal{D}), \ \ \theta \in \Theta. \tag{1}$$

To solve this optimization problem, we introduce Diff-BBO algorithm in Algorithm 1. At each iteration $k$, we train a conditional diffusion model and compute the optimal $y_k^*$ with the designed acquisition function. In practice, $y$ is selected from a constructed candidate set $\mathcal{Y}$ based on the acquisition function scores $\alpha(y)$. The range of $\mathcal{Y}$ is determined by $w \cdot \phi_k$, where $w$ is a positive scalar and $\phi_k$ is the maximum function values being queried in the current training dataset $\mathcal{D}$. Conditioning on $y_k^*$, we generate $N$ samples $\{\mathbf{x}_j\}_{j=1}^N$, where $\boldsymbol{x}_j \sim p_\theta(\mathbf{x} | y_k^*, \mathcal{D})$. By querying the oracle to evaluate $\boldsymbol{x}_j$, we obtain the best possible reconstructed value $\phi_k$ for the current iteration, and append all queried data pairs $\{\mathbf{x}_j, f(\mathbf{x}_j)\}_{j=1}^N$ to the training dataset $\mathcal{D}$. Figure 1 summarizes the difference between the prior forward BBO methods and our proposed inverse modeling approach.

## 2.1 Acquisition Function Design

In this section, we analyze the uncertainty of Diff-BBO, decomposing its uncertainty into aleatoric and epistemic uncertainty. Then we propose an acquisition function called Uncertainty-aware Exploration (UaE) based on it. We prove that by achieving a balance between high objective values and low epistemic uncertainty, UaE provides a near-optimal solution to the online BBO problem.

**Uncertainty Decomposition.**  We resort to the tools of Bayesian inference to solve the optimization problem defined in Equation (1). Given an observed value $y$ of a sample $\boldsymbol{x}$, the objective of Bayesian inference is to estimate the predictive distribution:

$$p(\boldsymbol{x} \mid y, \mathcal{D}) = \mathbb{E}_\theta[p_\theta(\boldsymbol{x} \mid y)] = \int_\theta p_\theta(\boldsymbol{x} \mid y) p(\theta \mid \mathcal{D}) d\theta. \tag{2}$$

By Equation (2), we recognize that the uncertainty arises from two sources: uncertainty in deciding parameter $\theta$ from its posterior $p(\theta|\mathcal{D})$ and uncertainty in generating sample $\boldsymbol{x}$ from a fixed diffusion model $p_\theta(\boldsymbol{x} \mid y)$ after $\theta$ is chosen. We further provide a decomposition in terms of the aleatoric uncertainty and its epistemic counterpart.

To estimate the aleatoric uncertainty, we can Monte Carlo (MC) sample $\boldsymbol{x}$ for $N$ times from a learned likelihood function $p_\theta(\boldsymbol{x} \mid y)$ for fixed $y, \theta$. To estimate the epistemic uncertainty, we use ensemble techniques. During the inference time, by initializing the trained ensemble models with different random seeds, we first sample $M$ model parameters $\{\theta_i\}_{i=1}^M$ to simulate $M$ conditional diffusion models. Then we generate $N$ samples $\{\boldsymbol{x}_j\}_{j=1}^N$ for each diffusion model with corresponding parameter $\theta_i$, $\forall i \in [M]$. Combining the above gives a practical way to decompose and estimate the two types of uncertainty, whish is formally described in Proposition 1.

**Proposition 1** (Uncertainty Decomposition). *At each iteration $k \in [K]$, the overall uncertainty in inverse modeling can be split into aleatoric and epistemic components, measured empirically as:*

$$\begin{aligned} \Delta_{aleatoric}(y, \mathcal{D}) &= \mathbb{E}_{\theta_i \sim p(\cdot|\mathcal{D})} \left[ \mathrm{Var}_{\boldsymbol{x}_{i,j} \sim p_{\theta_i}(\cdot|y)} \left( \|\boldsymbol{x}_{i,j}\| \right) \right], \ \ \forall i \in [M], j \in [N]; \\ \Delta_{epistemic}(y, \mathcal{D}) &= \mathrm{Var}_{\theta_i \sim p(\cdot|\mathcal{D})} \left( \mathbb{E}_{\boldsymbol{x}_{i,j} \sim p_{\theta_i}(\cdot|y)} \left[ \|\boldsymbol{x}_{i,j}\| \right] \right), \ \ \forall i \in [M], j \in [N]. \end{aligned} \tag{3}$$

**Uncertainty-aware Exploration.**  At each iteration $k \in [K]$ of Dif-BBO algorithm, the acquisition function proposes an optimal scalar value $y_k^*$ as follows: $y_k^* = \mathrm{argmax}_y \alpha(y, \mathcal{D})$, which is used to generate $\boldsymbol{x}$ in the design space using conditional difusion model.

Note that to design an effective acquisition function for inverse modeling, we need to achieve a balance between high objective values $y$ and low epistemic uncertainty. On the one hand, it is advantageous to focus on the regions in $\mathcal{X}$ whose corresponding $y$ is of high values. As function evaluations are expensive to perform, we prefer to generate samples $\boldsymbol{x}$ conditioned on higher $y$, and only query the oracle for such promising samples to solve the black-box optimization task. On the other hand, we employ the epistemic uncertainty to gauge the error in the trained diffusion model. Specifically, it helps reduce the approximation error between $y_k^*$ and the reconstructed function value $\max_{j \in [N]} f(\boldsymbol{x}_j)$, where $f(\cdot)$ is the black-box oracle, and $\boldsymbol{x}_j \sim p_\theta(\cdot|y_k^*, \mathcal{D}), \forall j \in [N]$.

We introduce the *Uncertainty-aware Exploration* (UaE) as our designed acquisition function:

$$\alpha(y, \mathcal{D}) = y - \Delta_{\text{epistemic}}(y, \mathcal{D}), \tag{4}$$

which utilizes the uncertainty estimation on conditional diffusion model as in Proposition 1. By balancing the exploration-exploitation trade-off, UaE effectively solve the online BBO problem.

**Performance Analyses of UaE.** We prove in Theorem 1 that by adopting UaE for inverse modeling to guide the selection of generated samples for solving BBO problems, we can obtain a near-optimal solution for the optimization problem defined in Equation (1). Detailed discussions and proofs can be found in Appendix C, Appendix D, and Appendix E.

**Theorem 1.** *Let $\mathcal{Y}$ be the constructed candidate set at each iteration $k \in [K]$ in Algorithm 1. By adopting UaE as the acquisition function to guide the sample generation process in conditional diffusion model, Diff-BBO (Algorithm 1) achieves a near-optimal solution for the online BBO problem defined in Equation (1):*

$$\max_{y_k \in \mathbb{R}} \sum_{k=1}^{K} f(\boldsymbol{x}_k), \ \ \boldsymbol{x}_k \sim p_\theta(\cdot \mid y_k, \mathcal{D}), \ \ \theta \in \Theta \ \ \Rightarrow \max_{y_k \in \mathcal{Y}} \sum_{k=1}^{K} \alpha(y_k, \mathcal{D}).$$

As a result, equipped with the novel design of UaE, Diff-BBO is a theoretically sound approach utilizing inverse modeling to effectievely solve the online BBO problem.

## 3 Experiments

To validate the efficacy of Diff-BBO, we conduct experiments on six real-world online black-box optimization tasks for both continuous and discrete optimization tasks.

**Dataset.** We restructured 5 real-world tasks from Design-Bench including 3 continuous and 2 discrete tasks. In **D'Kitty** and **Ant** Morphology, the goal is to optimize for the morphology of robots. In **Superconductor**, the aim is to optimize superconducting material with a high critical temperature. **TFBind8** and **TFBind10** are discrete tasks to find a DNA sequence with maximum affinity to bind with a specified transcription factor. We also include a **Molecular Discovery** task to optimize compound's activity against a biological target with therapeutic value. For each task, we arrange the offline dataset from Krishnamoorthy et al. (2023) in ascending order based on objective values and select data from the 25th to the 50th percentile as the initial training dataset. We prioritize data with lower objective scores to better observe performance differences across each baseline. Each optimization iteration is allocated 100 queries to the oracle function (batch size $N = 100$), with a total of 16 iterations conducted. More details of the dataset are provided in Appendix F.1.

**Baselines.** We compare Diff-BBO with 10 baselines, including Bayesian optimization (BO), trust region BO (TuRBO) (Eriksson et al., 2019), local latent space Bayesian optimization (LOL-BO) (Maus et al., 2022), likelihood-free BO (LFBO) (Song et al., 2022), evolutionary algorithms (Brindle, 1980; Real et al., 2019), conditioning by adaptive sampling (CbAS) (Brookes et al., 2019), and random sampling. For BO approaches, we include Gaussian Processes (GP) with Monte Carlo (MC)-based batch expected improvement (EI), MC-based batch upper confidence bound (UCB) (Wilson et al., 2017), and joint entropy search (JES (Hvarfner et al., 2022) as the acquisition functions. For LFBO, we use EI and probability of improvement (PI) as the acquisition functions.

**Results.** Figure 2 illustrates the performance across six datasets for all baselines and our proposed algorithm. Notably, Diff-BBO consistently outperforms other baselines in both discrete and continuous settings. Specifically, in the Ant and Dkitty tasks, Diff-BBO demonstrates a significant lead over all baseline methods, starting from the very first iteration of the online optimization process. This remarkable performance can be attributed to Diff-BBO's diffusion model-based inverse modeling approach, which effectively learns the data manifold in the design space from the initial dataset, even when the initial dataset lacks data with high objective function values.

## 4 Conclusion

In this paper, we introduced Diff-BBO, an inverse modeling approach for online black-box optimization that leverages the uncertainty of conditional diffusion models. By utilizing the novel acquisition function UaE, Diff-BBO strategically proposes objective function values to improve sample efficiency. We did extensive empirical evaluations to show the superior performance of Diff-BBO. Theoretically, we prove that using UaE leads to optimal optimization solutions.

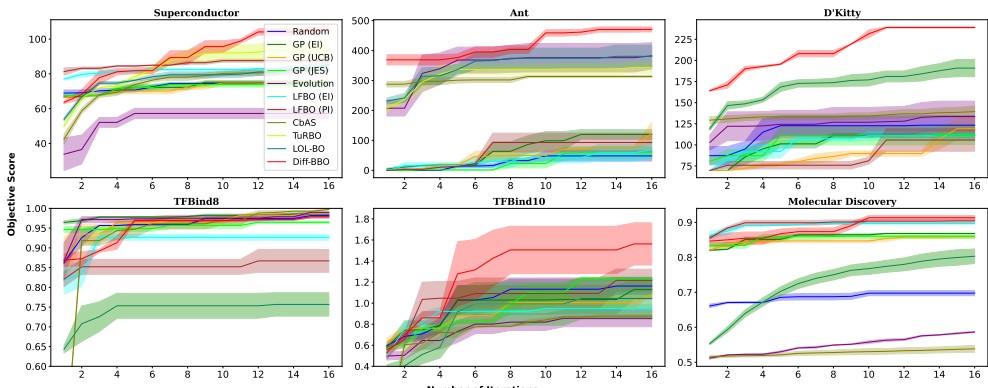

Figure 2: Comparison of Diff-BBO with baselines for online black-box optimization. We plot the mean and standard deviation across three random seeds.

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

# Appendices

## A  Diff-BBO algorithm

---

**Algorithm 1:** Diff-BBO

---

**Input:** Initial dataset $\mathcal{D} = \{\mathbf{x}, y\}$, total number of iterations $K$, candidate feasible range $C$,
oracle function $f(\cdot)$, batch size $N$

1 **Initialization**: Conditional diffusion model $p_\theta(\mathbf{x}|y)$
2 **for** $k = 1, 2, \cdots K$ **do**
3  $\quad$ Train the conditional diffusion model with $\mathcal{D}$
4  $\quad$ Construct a candidate set $\mathcal{Y} = \{y : 0 \leq y \leq C\}$
5  $\quad$ $y_k^* = \text{argmax}_{y \in \mathcal{Y}} \, \alpha(y, \mathcal{D})$
6  $\quad$ Generate $\{\mathbf{x}_j\}_{j=1}^N$ where $\mathbf{x}_j \sim p_\theta(\mathbf{x} \mid y_k^*, \mathcal{D})$
7  $\quad$ Query the oracle function $f(\cdot)$ with generated samples $\{\mathbf{x}_j\}_{j=1}^N$
8  $\quad$ $\mathcal{D} \leftarrow \mathcal{D} \cup \{\mathbf{x}_j, f(\mathbf{x}_j)\}_{j=1}^N$
9  $\quad$ $\phi_k \leftarrow \max(f(x)) \; s.t. \; x \in \mathcal{D}$
**Output:** Reconstructed $\{\phi_k\}_{k=1}^K$

---

## B  Conditional Diffusion Model Training

Diffusion Models (Sohl-Dickstein et al., 2015; Song et al., 2020) are probabilistic generative models that learn distributions through an iterative denoising process. These models consist of three components: a forward diffusion process that produces a series of noisy samples by adding Gaussian noise, a reverse process to reconstruct the original data samples from the noise, and a sampling procedure to generate new data samples from the learned distribution. Let the original sample be $\mathbf{x}_0$ and $t$ be the diffusion step. For conditional diffusion models, a conditional variable $y$ is added to both the forward process as $q(\boldsymbol{x}_t|\boldsymbol{x}_{t-1}, y)$ and reverse process as $p_\theta(\boldsymbol{x}_{t-1} \mid \boldsymbol{x}_t, y), \; \forall t \in [T]$.

The reverse process begins with the standard Gaussian distribution $p(\boldsymbol{x}_T) = \mathcal{N}(\mathbf{0}, \boldsymbol{I})$, and denoises $\boldsymbol{x}_t$ to recover $\boldsymbol{x}_0$ through the following Markov chain with reverse transitions:

$$p_\theta(\boldsymbol{x}_{0:T}|y) = p(\boldsymbol{x}_T) \prod_{t=1}^T p_\theta(\boldsymbol{x}_{t-1} \mid \boldsymbol{x}_t, y), \;\; \boldsymbol{x}_T \sim \mathcal{N}(\mathbf{0}, \boldsymbol{I}),$$

$$p_\theta(\boldsymbol{x}_{t-1} \mid \boldsymbol{x}_t, y) = \mathcal{N}(\boldsymbol{x}_{t-1}; \mu_\theta(\boldsymbol{x}_t, t, y), \Sigma_\theta(\boldsymbol{x}_t, t, y)).$$

During training, $\Sigma_\theta$ is empirically fixed, and $\mu_\theta$ is reparametrized by a trainable denoise function $\boldsymbol{\epsilon}_\theta(\boldsymbol{x}_t, t, y)$, which is used to estimate the noise vector $\epsilon$ that was added to input $\boldsymbol{x}_t$, and is trained by minimizing a reweighted version of the evidence lower bound (ELBO):

$$\mathcal{L}_{\text{dif}} = \mathbb{E}_{\boldsymbol{x}_0 \sim q(\boldsymbol{x}), y, \boldsymbol{\epsilon} \sim \mathcal{N}(0, \boldsymbol{I}), t \sim \mathcal{U}(0, T), \boldsymbol{x}_t \sim q(\boldsymbol{x}_t | \boldsymbol{x}_0, y)} \left[ w(t) \| \boldsymbol{\epsilon} - \boldsymbol{\epsilon}_\theta(\boldsymbol{x}_t, t, y) \|_2^2 \right]. \tag{5}$$

Note that the loss in Equation (5) (Ho et al., 2020) for $\boldsymbol{\epsilon}_\theta$ is denoising score matching for all time step $t$, which estimates the gradient of the log probability density of the noisy data (a.k.a. score function): $\boldsymbol{\epsilon}_\theta(\boldsymbol{x}_t, t, y) \approx -\sigma_t \nabla_{\boldsymbol{x}} \log p(\boldsymbol{x} \mid y)$. We further denote the score function as $s_\theta(\boldsymbol{x}_t, y, t) := -\boldsymbol{\epsilon}_\theta(\boldsymbol{x}_t, t, y) / \sigma_t$.

Instead of learning a fixed deterministic $\theta$ from a deterministic neural network, we are interested in learning its Bayesian posterior to further understand and improve the model's performance as well as its reliability with uncertainty quantification. In Bayesian settings, we consider the model parameters $\theta \in \Theta$, where $\Theta$ is the parameter space, and maintain its posterior distribution $p(\theta|\mathcal{D})$, which is learned from training data $\mathcal{D}$. By choosing $\theta$ from its posterior, essentially we sample a score function $\widetilde{s}_\theta(\boldsymbol{x}_t, y, t)$ from the probability distribution $p(s_\theta \mid \boldsymbol{x}_t, y, t, \mathcal{D}) = \mathcal{N}(s_\theta(\boldsymbol{x}_t, y, t), \Sigma_{s_\theta}(\boldsymbol{x}_t, y, t))$, whose expected value is $s_\theta(\boldsymbol{x}_t, y, t)$, and variance is a diagonal covariance matrix $\Sigma_{s_\theta}(\boldsymbol{x}_t, y, t)$.

Specifically, we adopt classifier-free guidance as in (Ho and Salimans, 2022) to eliminate the requirement of training a separate classifier model. We jointly train an unconditional diffusion model $p_\theta(\boldsymbol{x})$ parameterized by $\epsilon_\theta(\mathbf{x}, t, \emptyset)$ and a conditional diffusion model $p_\theta(\boldsymbol{x}|y)$ parameterized by $\epsilon_\theta(\mathbf{x}, t, y)$ by minimizing the following loss function:

$$\mathcal{L}_{\text{cdif}} = \mathbb{E}_{\boldsymbol{x}_0, y, \boldsymbol{\epsilon}, t, \boldsymbol{x}_t, \lambda} \left[ w(t) \| \boldsymbol{\epsilon} - \boldsymbol{\epsilon}_\theta(\boldsymbol{x}_t, t, (1-\lambda)y + \lambda\emptyset) \|_2^2 \right], \tag{6}$$

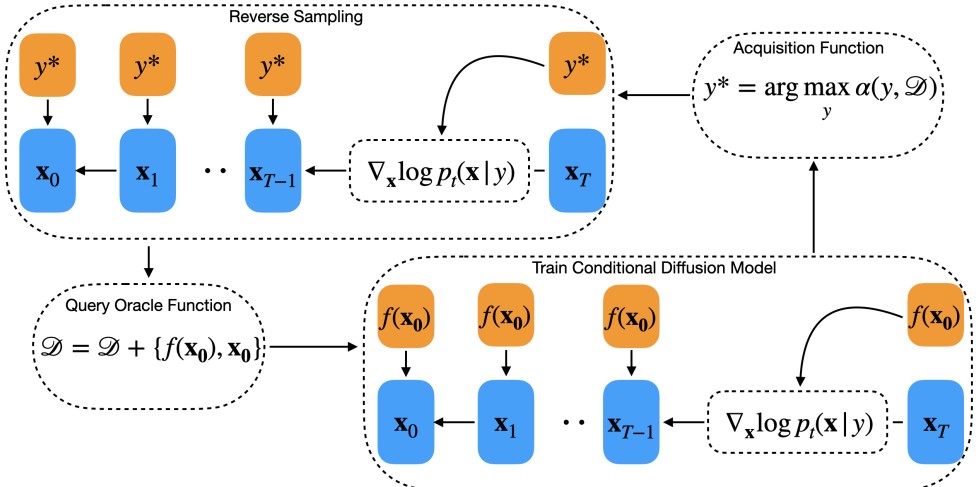

Figure 3: Black-box optimization framework using the conditional diffusion model as the inverse model. The overall framework includes 4 stages. 1. Train the conditional diffusion model given the current training dataset. 2. Compute the acquisition function and select the optimal $y^*$ to condition on. 3. Generate samples $\{\mathbf{x_0}\}$ conditioned on $y^*$. 4. Query the oracle given generated samples $\{\mathbf{x_0}\}$ and update the training dataset.

where $\boldsymbol{x}_0 \sim q(\boldsymbol{x}), \boldsymbol{\epsilon} \sim \mathcal{N}(0, \boldsymbol{I}), t \sim \mathcal{U}(0, T), \boldsymbol{x}_t \sim q(\boldsymbol{x}_t \mid \boldsymbol{x}_0), \lambda \sim \text{Bernoulli}(p_{\text{uncond}})$, and $p_{\text{uncond}}$ is the probability of setting $y$ to the unconditional information $\emptyset$. The overall Diff-BBO framework with the conditional diffusion model included in shown in Figure 3.

## C   Uncertainty Quantification on Conditional Diffusion Model

Here, let us first consider the problem of how to capture the uncertainty for a fixed diffusion model. In fact, the uncertainty in generating $\boldsymbol{x}$ can be explicitly traced through the denoising process. More specifically, Theorem 2 provides analytical solutions to compute the uncertainty on a single denoising process of general score-based conditional diffusional models. It provides theoretical insights of how uncertainty is being propagated through the reverse denoising process both in discrete time and continuous time, which is characterized through the lens of stochastic differential equations (SDEs) of the Ornstein–Uhlenbeck (OU) process.

**Theorem 2.** *(Uncertainty propagation) Let $t \in [T]$ be the diffusion step, $s_\theta(\boldsymbol{x}, y, t)$ be the score function of the corresponding diffusion model $p_\theta(\boldsymbol{x} \mid y)$. For a single conditional diffusional model $p_\theta(\boldsymbol{x} \mid y)$, the uncertainty in generating a sample $\boldsymbol{x}$ can be analytically traced through the discrete-time reverse denoising process as follows:*

$$\text{Var}(\boldsymbol{x}_{t-1}) = \tfrac{1}{4}\text{Var}(\boldsymbol{x}_t) + \text{Var}(s_\theta(\boldsymbol{x}, y, t)) + \tfrac{1}{2}\left(\mathbb{E}\left[\boldsymbol{x}_t \circ s_\theta(\boldsymbol{x}_t, y, t)\right] - \mathbb{E}[\boldsymbol{x}_t] \circ \mathbb{E}[s_\theta(\boldsymbol{x}_t, y, t)]\right) + I,$$

$$\mathbb{E}(\boldsymbol{x}_{t-1}) = \frac{1}{2}\mathbb{E}(\boldsymbol{x}_t) + \mathbb{E}(s_\theta(\boldsymbol{x}, y, t)),$$

*where $\circ$ is the Hadamard product, and $I$ is the identity matrix. Similarly, in continuous-time process, the uncertainty can be captured as follows:*

$$\text{Var}(\boldsymbol{x}_0) = (T + 1)I + \text{Var}\left(\int_{t=0}^{T}\left(\frac{1}{2}\boldsymbol{x}_t + s_\theta(\boldsymbol{x}, y, t)\right) \mathrm{d}t\right). \tag{7}$$

Nevertheless, performing exact Bayesian inference for uncertainty quantification when training diffusion models requires non-trivial efforts and can be computationally demanding. Hence, we introduce a practically-efficient uncertainty decomposition based on Equation (2).

### C.1   Conditional Diffusion SDE

It can be shown that the conditional diffusion model can be represented by the Ornstein–Uhlenbeck (OU) process, which is a time-homogeneous continuous-time Markov process:

$$\mathrm{d}\boldsymbol{x}_t = -\gamma\boldsymbol{x}_t \,\mathrm{d}t + \sigma \,\mathrm{d}\boldsymbol{w}_t, \tag{8}$$

where $\gamma$ is the relaxation rate, $\sigma$ is the strength of fluctuation, and $\boldsymbol{w}_t$ is the standard Wiener process (a.k.a., Brownian motion). Both $\gamma$ and $\sigma$ are time-invariant. In particular, setting $\gamma = 1$ and $\sigma = \sqrt{2}$, we are able to establish that Denoising Diffusion Probabilistic Model (DDPM) is equivalent to OU process observed at discrete times. In the remaining text, we consider SDEs for general score-based diffusion models. The SDE of the forward process in conditional diffusion model can then be written as:

$$\mathrm{d}\boldsymbol{x}_t = -\frac{1}{2}g(t)\boldsymbol{x}_t\,\mathrm{d}t + \sqrt{g(t)}\,\mathrm{d}\boldsymbol{w}_t, \quad \boldsymbol{x}_0 \sim q(\boldsymbol{x}|y) \tag{9}$$

where $g(t)$ is a nondecreasing weighting function that controls the speed of diffusion in the forward process and $g(t) > 0$. For simplicity of analysis, we fix $g(t) = 1$ for all $t \in [T]$.

The generation process of a conditional score-based diffusion model can be viewed as a particular discretization of the following reverse-time SDE:

$$\mathrm{d}\boldsymbol{x}_t = \left(\frac{1}{2}\boldsymbol{x}_t - \nabla_{\boldsymbol{x}_t} \log p(\boldsymbol{x}_t|y)\right)\mathrm{d}t + \mathrm{d}\boldsymbol{w}_t, \quad \boldsymbol{x}_0 \sim p(\boldsymbol{x}_T|y). \tag{10}$$

In practice, the unknown ground truth conditional score $\nabla_{\boldsymbol{x}_t} \log p(\boldsymbol{x}_t|y)$ needs to be estimated with score networks. Let such estimator denoted by $s_\theta(\boldsymbol{x}, y, t)$, then the conditional sample generation is to simulate the following backward SDE:

$$\mathrm{d}\boldsymbol{x}_t = \left(\frac{1}{2}\boldsymbol{x}_t - s_\theta(\boldsymbol{x}, y, t)\right)\mathrm{d}t + \mathrm{d}\boldsymbol{w}_t, \quad \boldsymbol{x}_0 \sim \mathcal{N}(\boldsymbol{0}, \boldsymbol{I}). \tag{11}$$

In Bayesian settings, we sample a score function $\widetilde{s}_\theta(\boldsymbol{x}_t, y, t)$ from the probability distribution $p(s_\theta|\boldsymbol{x}_t, y, t, \mathcal{D}) = \mathcal{N}(s_\theta(\boldsymbol{x}_t, y, t), \Sigma_\theta(\boldsymbol{x}_t, y, t))$ with expected value $s_\theta(\boldsymbol{x}_t, y, t)$, and diagonal covariance $\Sigma_\theta(\boldsymbol{x}_t, y, t)$.

### C.2 Estimation of Uncertainty

In this section, we quantify the uncertainty of a single conditional diffusion model in both discrete-time and continuous-time reverse process for Theorem 2.

#### C.2.1 Uncertainty in Discrete-time Reverse Process

We first proof the first statement of Theorem 2. We consider the Euler discretization of Equation (11), which leads to:

$$\boldsymbol{x}_{t-1} = \frac{1}{2}\boldsymbol{x}_t + s_\theta(\boldsymbol{x}, y, t) + \boldsymbol{\epsilon}, \quad \boldsymbol{\epsilon} \sim \mathcal{N}(\boldsymbol{0}, \boldsymbol{I}). \tag{12}$$

We thus have,

$$\mathrm{Var}(\boldsymbol{x}_{t-1}) = \frac{1}{4}\mathrm{Var}(\boldsymbol{x}_t) + \mathrm{Var}(s_\theta(\boldsymbol{x}, y, t)) + \frac{1}{2}\mathrm{Cov}\left(\boldsymbol{x}_t, s_\theta(\boldsymbol{x}, y, t)\right) + I. \tag{13}$$

$$\mathbb{E}(\boldsymbol{x}_{t-1}) = \frac{1}{2}\mathbb{E}(\boldsymbol{x}_t) + \mathbb{E}(s_\theta(\boldsymbol{x}, y, t)). \tag{14}$$

Here $\mathrm{Cov}\left(\boldsymbol{x}_t, s_\theta(\boldsymbol{x}, y, t)\right)$ is the element-vise covariance between $\boldsymbol{x}_t$ and $s_\theta(\boldsymbol{x}, y, t)$. Note that we only need to consider the correlation between $\boldsymbol{x}_t$ and $s_\theta(\boldsymbol{x}, y, t)$ at the same time step. As a result, to estimate $\mathrm{Cov}\left(\boldsymbol{x}_t, s_\theta(\boldsymbol{x}, y, t)\right)$, we have,

$$\begin{aligned}
\mathrm{Cov}\left(\boldsymbol{x}_t, s_\theta(\boldsymbol{x}, y, t)\right) &= \mathbb{E}\left[\left(\boldsymbol{x}_t - \mathbb{E}[\boldsymbol{x}_t]\right)\left(s_\theta(\boldsymbol{x}, y, t) - \mathbb{E}[s_\theta(\boldsymbol{x}, y, t)]\right)^{\mathrm{T}}\right] \\
&= \mathbb{E}\left[\boldsymbol{x}_t \circ s_\theta(\boldsymbol{x}, y, t)\right] - \mathbb{E}[\boldsymbol{x}_t] \circ \mathbb{E}[s_\theta(\boldsymbol{x}, y, t)] \\
&= \mathbb{E}_{\boldsymbol{x}_t}\left[\boldsymbol{x}_t \circ s_\theta(\boldsymbol{x}, y, t)\right] - \mathbb{E}[\boldsymbol{x}_t] \circ \mathbb{E}_{\boldsymbol{x}_t}[s_\theta(\boldsymbol{x}_t, y, t)]
\end{aligned} \tag{15}$$

where $\circ$ is the Hadamard product and the third equality is by tower's rule. Substituting Equation (15) back to Equation (13) completes the proof of the first part of Theorem 2.

#### C.2.2 Uncertainty in Continuous-time Reverse Process

We now proof the second statement of Theorem 2. To perform the uncertainty quantification for the continuous-time reverse process, we posit the following assumption.

**Assumption 1.** *For valid $t \in [0, T]$, the generating process $\boldsymbol{x}_t$ in Equation (10) is integrable and has finite second-order moments.*

With Assumption 1, integrating Equation (10) with respect to $t$ yields:

$$\boldsymbol{x}_0 = \boldsymbol{x}_T - \int_{t=0}^{T} \left( \frac{1}{2}\boldsymbol{x}_t + \nabla_{\boldsymbol{x}_t} \log p(\boldsymbol{x}_t|y) \right) \mathrm{d}t + \int_{t=0}^{T} \mathrm{d}\boldsymbol{w}_t. \tag{16}$$

Applying the variance operator to both sides of

$$\mathrm{Var}(\boldsymbol{x}_0) = \mathrm{Var}(\boldsymbol{x}_T) + \mathrm{Var}\left( \int_{t=0}^{T} \left( \frac{1}{2}\boldsymbol{x}_t + \nabla_{\boldsymbol{x}_t} \log p(\boldsymbol{x}_t|y) \right) \mathrm{d}t \right) + \mathrm{Var}\left( \int_{t=0}^{T} \mathrm{d}\boldsymbol{w}_t \right)$$

$$= I + \mathrm{Var}\left( \int_{t=0}^{T} \left( \frac{1}{2}\boldsymbol{x}_t + \nabla_{\boldsymbol{x}_t} \log p(\boldsymbol{x}_t|y) \right) \mathrm{d}t \right) + \mathbb{E}\left[ \left( \int_{t=0}^{T} \mathrm{d}\boldsymbol{w}_t \right)^2 \right] - \left( \mathbb{E}\left[ \int_{t=0}^{T} \mathrm{d}\boldsymbol{w}_t \right] \right)^2$$

$$= (T+1)I + \underbrace{\mathrm{Var}\left( \int_{t=0}^{T} \left( \frac{1}{2}\boldsymbol{x}_t + \nabla_{\boldsymbol{x}_t} \log p(\boldsymbol{x}_t|y) \right) \mathrm{d}t \right)}_{V_1}, \tag{17}$$

where the last equality follows the properties of Itô Integral and rules of stochastic calculus such that $(\mathrm{d}\boldsymbol{w})^2 = \mathrm{d}t$, $\mathbb{E}[\int_{t=0}^{T} \mathrm{d}\boldsymbol{w}_t] = 0$. Hence, to provide an uncertainty estimate for $\boldsymbol{x}_0$, it remains to estimate the term $V_1$. Recall that the true score function $\nabla_{\boldsymbol{x}_t} \log p(\boldsymbol{x}_t|y)$ is approximated by $s_\theta((\boldsymbol{x}_t, y, t) = -\boldsymbol{\epsilon}_\theta(\boldsymbol{x}_t, t, y)/\sigma_t$. For ease of notation, let $s_{\theta,t} = s_\theta(\boldsymbol{x}_t, y, t)$ and $\widetilde{s}_{\theta,t} = \widetilde{s}_\theta(\boldsymbol{x}_t, y, t)$, which gives

$$V_1 = \int_{t=0}^{T} \int_{s=0}^{T} \left( \frac{1}{4}\mathrm{Cov}(\boldsymbol{x}_s, \boldsymbol{x}_t) - \frac{1}{2}\mathrm{Cov}(\boldsymbol{x}_s, s_{\theta,t}) - \frac{1}{2}\mathrm{Cov}(\boldsymbol{x}_t, s_{\theta,s}) + \mathrm{Cov}(s_{\theta,t}, s_{\theta,s}) \right) \mathrm{d}s\,\mathrm{d}t.$$

When $s \neq t$, score functions $s_{\theta,t}$ and $s_{\theta,s}$ are independent, and similarly, $\boldsymbol{x}_t$ and $s_{\theta,s}$ are also independent. As a result, the above equation can be further simplified as

$$V_1 = \int_{t=0}^{T} \int_{s=0}^{T} \left( \frac{1}{4}\mathrm{Cov}(\boldsymbol{x}_s, \boldsymbol{x}_t) - \frac{1}{2}\mathrm{Cov}(\boldsymbol{x}_s, s_{\theta,t}) \right) \mathrm{d}s\,\mathrm{d}t - \int_{t=0}^{T} \left( \mathrm{Cov}(\boldsymbol{x}_t, s_{\theta,t}) + \mathrm{Cov}(s_{\theta,t}, s_{\theta,t}) \right) \mathrm{d}t.$$

Combining all the above results together completes the proof of the second statement of Theorem 2.

## D  Performance Analysis of UaE

To quantify the quality of generated samples, we theoretically analyze the sub-optimality performance gap between $y_k^*$ and reconstructed value at each iteration. In particular, Theorem 3 and Theorem 4 demonstrate that such sub-optimality gap can be effectively handled in inverse modeling, with proofs deferred to Appendix D.1. We first show that by using conditional diffusion model, the expected error of the sub-optimality performance gap is zero.

**Theorem 3.** *At each iteration $k \in [K]$, define the sub-optimality performance gap as*

$$\Delta(p_\theta, y_k^*) = \left| y_k^* - \max_{j \in [N]} f(\boldsymbol{x}_j) \right|, \quad \text{where } \boldsymbol{x}_j \sim p_\theta(\cdot|y_k^*, \mathcal{D}), \ \forall j \in [N]. \tag{18}$$

*Assume that there exists some $\theta^* \sim p(\theta|\mathcal{D})$ that produces a predictive distribution $p_{\theta^*}(\cdot \mid \mathcal{D})$ such that it is able to generate a sample $\boldsymbol{x}^*$ that perfectly reconstructs $y_k^*$. Suppose function $f$ is L-Lipschitz and each sample is $\sigma$-subGaussian, it can be shown that*

$$\mathbb{E}\left[ \Delta(p_\theta, y_k^*) \right] \leq c_1 L \sqrt{d}\sigma,$$

*where $c_1$ is some universal constant and the empirical estimator $\widehat{\mathbb{E}}\left[ \Delta(p_\theta, y_k^*) \right]$ is unbiased.*

Theorem 3 suggests that in expectation, the reconstructed function value $\max_{j \in [N]} f(\boldsymbol{x}_j)$ is able to accurately recover the provided conditional information $y_k^*$. Hence, in order to obtain a reasonable estimator for the optimization problem, the remaining concern goes to the variance of the gap defined in Equation (18), which is further evaluated in Theorem 4.

**Theorem 4.** *(Sub-optimality bound) At each iteration $k \in [K]$, suppose $M$ model parameters $\{\theta_i\}_{i=1}^M$ are generated from the ensemble model for some fixed dataset $\mathcal{D}$. Suppose function $f$ is $L$-Lipschitz, it can be shown that the variance of the sub-optimality performance gap of each model is bounded by the epidemic uncertainty:*

$$\text{Var}\left(\Delta(p_{\theta_i}, y_k^*)\right) \leq c_2 L^2 d\sigma^2 + c_2 L^2 \Delta_{\text{epistemic}}(y_k^*, \mathcal{D}), \quad \forall i \in M, \tag{19}$$

*where $c_2$ is some universal positive constant.*

Theorem 4 shows that the variance of the sub-optimality performance gap can be upper bounded by the epistemic uncertainty of diffusion model. Therefore, our proposed acquisition function achieves the balance between high objective value and low epistemic uncertainty.

### D.1 Analysis of Sub-optimality

In this section, we study the behavior of the sub-optimality gap of our algorithm by proving Theorem 3 and Theorem 4. We first introduce the notation that is used throughout this section and the next section. Then we present the main lemmas along with their proofs. Finally, we combine the lemmas to prove our main results.

At each iteration $k \in [K]$, let $y_k^*$ be the target function value on which the diffusion model conditions, and $p_\theta$ be the model learned by the conditional diffusion model. We define the performance metric for online BBO problem, which measures the sub-optimal performance gap between the function value achieved by sample $\boldsymbol{x} \sim p_\theta(\cdot|y_k^*, \mathcal{D})$ and the target function value $y_k^*$. Its formal definition is described as follows:

$$\Delta(p_\theta, y_k^*) = \left| y_k^* - \max_{j \in [N]} f(\boldsymbol{x}_j) \right|, \quad \text{where } \boldsymbol{x}_j \sim p_\theta(\cdot|y_k^*, \mathcal{D}), \ \forall j \in [N]. \tag{20}$$

For simplicity of analysis, we consider $N = 1$, and let the generated sample at the $k$-th iteration be $\boldsymbol{x}_k$ in the remaining text. We remark that all proofs go through smoothly for general $N$ with more nuanced notations, and do not affect the conclusions being drawn. To proceed with the proofs in this section, we first state the formal assumptions for the black-box function $f(\cdot)$ and sample $\boldsymbol{x}$.

**Assumption 2.** *The scalar black-box function $f$ is $L$-Lipschitz in $\boldsymbol{x}$:*

$$|f(\boldsymbol{x}') - f(\boldsymbol{x})| \leq L\|\boldsymbol{x}' - \boldsymbol{x}\|, \quad \forall \boldsymbol{x}', \boldsymbol{x} \in \mathbb{R}^d.$$

**Assumption 3.** *Each generated sample $\boldsymbol{x} \in \mathbb{R}^d$ is $\sigma$-subGaussian. That is, there exists $\sigma \in \mathbb{R}$ such that for any $\boldsymbol{v} \in \mathbb{R}^d$ with $\|\boldsymbol{v}\| = 1$, $\boldsymbol{v}^{\text{T}}(\boldsymbol{x} - \mathbb{E}[\boldsymbol{x}])$ is $\sigma$-subGaussian, and its moment generating function is bounded by:*

$$\mathbb{E}[\exp\left(\lambda \boldsymbol{v}^{\text{T}}(\boldsymbol{x} - \mathbb{E}[\boldsymbol{x}])\right)] \leq \exp\left(\frac{\sigma^2 \lambda^2}{2}\right), \quad \forall \lambda \in \mathbb{R}, \ \boldsymbol{v} \in \mathbb{S}^{d-1},$$

*where $\mathbb{S} := \{\boldsymbol{v} \in \mathbb{R}^d : \|\boldsymbol{v}\| = 1\}$ is the $(d-1)$ unit sphere.*

Before proceeding with the proofs of main theorems, we present our main lemmas.

**Lemma D.1.** *At each iteration $k \in [K]$, under fixed parameters $\theta$ and $\theta^*$, for $\boldsymbol{x}_k \sim p_\theta(\cdot|y_k^*, \mathcal{D})$, $\boldsymbol{x}^* \sim p_{\theta^*}(\cdot|y_k^*, \mathcal{D})$, we have*

$$\mathbb{E}_{\boldsymbol{x}_k \sim p_\theta(\cdot|y_k^*, \mathcal{D}), \boldsymbol{x}^* \sim p_{\theta^*}(\cdot|y_k^*, \mathcal{D})} \left[\|\boldsymbol{x}^* - \boldsymbol{x}_k\|\right] \leq 8\sqrt{d}\sigma + \|\mathbb{E}_{\boldsymbol{x}^*}[\boldsymbol{x}^*] - \mathbb{E}_{\boldsymbol{x}_k}[\boldsymbol{x}_k]\|, \tag{21}$$

$$\mathbb{E}_{\boldsymbol{x}_k \sim p_\theta(\cdot|y_k^*, \mathcal{D}), \boldsymbol{x}^* \sim p_{\theta^*}(\cdot|y_k^*, \mathcal{D})} \left[\|\boldsymbol{x}^* - \boldsymbol{x}_k\|\right] \geq \|\mathbb{E}_{\boldsymbol{x}^*}[\boldsymbol{x}^*] - \mathbb{E}_{\boldsymbol{x}_k}[\boldsymbol{x}_k]\|. \tag{22}$$

*Proof of Lemma D.1.* To bound $\mathbb{E}\left[\|\boldsymbol{x}^* - \boldsymbol{x}_k\|\right]$, by triangle inequality,

$$\mathbb{E}_{\boldsymbol{x}_k, \boldsymbol{x}^*}\left[\|\boldsymbol{x}^* - \boldsymbol{x}_k\|\right] = \mathbb{E}\left[\|\boldsymbol{x}^* - \mathbb{E}[\boldsymbol{x}^*] + \mathbb{E}[\boldsymbol{x}_k] - \boldsymbol{x}_k + \mathbb{E}[\boldsymbol{x}^*] - \mathbb{E}[\boldsymbol{x}_k]\|\right]$$

$$\leq \mathbb{E}\left[\|\boldsymbol{x}^* - \mathbb{E}[\boldsymbol{x}^*]\|\right] + \mathbb{E}\left[\|\boldsymbol{x}_k - \mathbb{E}[\boldsymbol{x}_k]\|\right] + \mathbb{E}\left[\|\mathbb{E}[\boldsymbol{x}^*] - \mathbb{E}[\boldsymbol{x}_k]\|\right].$$

Under assumption 3, by Lemma D.3, we have,

$$\mathbb{E}_{\boldsymbol{x}_k, \boldsymbol{x}^*}\left[\|\boldsymbol{x}^* - \boldsymbol{x}_k\|\right] \leq 8\sqrt{d}\sigma + \|\mathbb{E}[\boldsymbol{x}^*] - \mathbb{E}[\boldsymbol{x}_k]\|.$$

Applying triangle inequality completes the step. In addition, it can be easily seen that

$$\mathbb{E}_{\boldsymbol{x}_k, \boldsymbol{x}^*}\left[\|\boldsymbol{x}^* - \boldsymbol{x}_k\|\right] \geq \|\mathbb{E}[\boldsymbol{x}^*] - \mathbb{E}[\boldsymbol{x}_k]\|.$$

$\square$

**Lemma D.2.** *At each iteration $k \in [K]$, under fixed parameters $\theta$ and $\theta^*$, for $\boldsymbol{x}_k \sim p_\theta(\cdot|y_k^*, \mathcal{D})$, $\boldsymbol{x}^* \sim p_{\theta^*}(\cdot|y_k^*, \mathcal{D})$, we have*

$$\mathrm{Var}_{\boldsymbol{x}_k \sim p_\theta(\cdot|y_k^*, \mathcal{D}), \boldsymbol{x}^* \sim p_{\theta^*}(\cdot|y_k^*, \mathcal{D})}(\|\boldsymbol{x}^* - \boldsymbol{x}_k\|) \leq c_3 d\sigma^2. \tag{23}$$

*Proof of Lemma D.2.* By definition of variance,

$$\mathrm{Var}_{\boldsymbol{x}_k, \boldsymbol{x}^*}(\|\boldsymbol{x}^* - \boldsymbol{x}_k\|) = \mathbb{E}[\|\boldsymbol{x}^* - \boldsymbol{x}_k\|^2] - (\mathbb{E}[\|\boldsymbol{x}^* - \boldsymbol{x}_k\|])^2. \tag{24}$$

Expanding the first term leads to

$$
\begin{aligned}
\mathbb{E}[\|\boldsymbol{x}^* - \boldsymbol{x}_k\|^2] &= \mathbb{E}[(\boldsymbol{x}^* - \boldsymbol{x}_k)^{\mathrm{T}}(\boldsymbol{x}^* - \boldsymbol{x}_k)] \\
&= \mathbb{E}[\|\boldsymbol{x}^*\|^2] + \mathbb{E}[\|\boldsymbol{x}_k\|^2] - 2\mathbb{E}[(\boldsymbol{x}_k)^{\mathrm{T}}\boldsymbol{x}^*] \\
&= \mathbb{E}[\|\boldsymbol{x}^*\|^2] + \mathbb{E}[\|\boldsymbol{x}_k\|^2] - 2\mathbb{E}[(\boldsymbol{x}_k)]^{\mathrm{T}}\mathbb{E}[\boldsymbol{x}^*],
\end{aligned} \tag{25}
$$

where the last equality is due to the independece between $\boldsymbol{x}^*$ and $\boldsymbol{x}_k$.

Under Assumption 3 and by Lemma D.4, we have

$$
\begin{aligned}
\mathbb{E}[\|\boldsymbol{x}^*\|^2] &= \mathbb{E}[\|\boldsymbol{x}^* - \mathbb{E}[\boldsymbol{x}^*] + \mathbb{E}[\boldsymbol{x}^*]\|^2] \\
&= \mathbb{E}[(\boldsymbol{x}^* - \mathbb{E}[\boldsymbol{x}^*])^{\mathrm{T}}(\boldsymbol{x}^* - \mathbb{E}[\boldsymbol{x}^*])] + \|\mathbb{E}[\boldsymbol{x}^*]\|^2 \\
&= \mathrm{tr}(\mathbb{E}[(\boldsymbol{x}^* - \mathbb{E}[\boldsymbol{x}^*])(\boldsymbol{x}^* - \mathbb{E}[\boldsymbol{x}^*])]^{\mathrm{T}}) + \|\mathbb{E}[\boldsymbol{x}^*]\|^2 \\
&\leq Cd\sigma^2 + \|\mathbb{E}[\boldsymbol{x}^*]\|^2.
\end{aligned}
$$

Here, the second equality holds as the cross terms vanish due to the fact that $\mathbb{E}[\boldsymbol{x}^* - \mathbb{E}[\boldsymbol{x}^*]] = 0$. Similarly,

$$\mathbb{E}[\|\boldsymbol{x}_k\|^2] \leq Cd\sigma^2 + \|\mathbb{E}[\boldsymbol{x}_k]\|^2.$$

Substituting the above two results back to Equation (25),

$$
\begin{aligned}
\mathbb{E}[\|\boldsymbol{x}^* - \boldsymbol{x}_k\|^2] &\leq 2Cd\sigma^2 + \|\mathbb{E}[\boldsymbol{x}_k]\|^2 + \|\mathbb{E}[\boldsymbol{x}^*]\|^2 - 2\mathbb{E}[(\boldsymbol{x}_k)^{\mathrm{T}}\boldsymbol{x}^*] \\
&\leq 2Cd\sigma^2 + \|\mathbb{E}[\boldsymbol{x}_k] - \mathbb{E}[\boldsymbol{x}^*]\|^2.
\end{aligned} \tag{26}
$$

Substituting Equation (26) back to Equation (24) and applying Lemma D.1 leads to

$$\mathrm{Var}_{\boldsymbol{x}_k, \boldsymbol{x}^*}(\|\boldsymbol{x}^* - \boldsymbol{x}_k\|) \leq 2Cd\sigma^2 + \|\mathbb{E}[\boldsymbol{x}_k] - \mathbb{E}[\boldsymbol{x}^*]\|^2 - (8\sqrt{d}\sigma + \|\mathbb{E}[\boldsymbol{x}^*] - \mathbb{E}[\boldsymbol{x}_k]\|)^2 \leq c_3 d\sigma^2.$$

$\square$

With the above results, we are ready to prove Theorem 3 and Theorem 4.

**Theorem 3.** *At each iteration $k \in [K]$, define the sub-optimality performance gap as*

$$\Delta(p_\theta, y_k^*) = \left| y_k^* - \max_{j \in [N]} f(\boldsymbol{x}_j) \right|, \quad \text{where} \ \boldsymbol{x}_j \sim p_\theta(\cdot|y_k^*, \mathcal{D}), \ \forall j \in [N]. \tag{18}$$

*Assume that there exists some $\theta^* \sim p(\theta|\mathcal{D})$ that produces a predictive distribution $p_{\theta^*}(\cdot \mid \mathcal{D})$ such that it is able to generate a sample $\boldsymbol{x}^*$ that perfectly reconstructs $y_k^*$. Suppose function $f$ is $L$-Lipschitz and each sample is $\sigma$-subGaussian, it can be shown that*

$$\mathbb{E}[\Delta(p_\theta, y_k^*)] \leq c_1 L\sqrt{d}\sigma,$$

*where $c_1$ is some universal constant and the empirical estimator $\widehat{\mathbb{E}}[\Delta(p_\theta, y_k^*)]$ is unbiased.*

*Proof of Theorem 3.* Recall that we consider the case where $N = 1$, and denote $\boldsymbol{x}_k$ the generated sample in the $k$-th iteration, i.e. $\boldsymbol{x}_k \sim p_\theta(\cdot|y_k^*, \mathcal{D})$, where $\theta \sim p(\theta \mid \mathcal{D})$. In each iteration $k$, with the existence of $\theta^* \sim p(\theta \mid \mathcal{D})$, we have $y_k^* = f(\boldsymbol{x}^*)$, where $\boldsymbol{x}^* \sim p_{\theta^*}(\cdot|y_k^*, \mathcal{D})$. Hence, under Assumption 2,

$$\mathbb{E}[\Delta(p_\theta, y_k^*)] = \mathbb{E}[|f(\boldsymbol{x}^*) - f(\boldsymbol{x}_k)|] \leq L\mathbb{E}[\|\boldsymbol{x}^* - \boldsymbol{x}_k\|].$$

By Lemma D.1, we have

$$\mathbb{E}[\Delta(p_\theta, y_k^*)] \leq 8L\sqrt{d}\sigma + \mathbb{E}_{\theta, \theta^*}[\|\mathbb{E}[\boldsymbol{x}^*] - \mathbb{E}[\boldsymbol{x}_k]\|].$$

$\square$

**Theorem 4.** *(Sub-optimality bound) At each iteration $k \in [K]$, suppose $M$ model parameters $\{\theta_i\}_{i=1}^M$ are generated from the ensemble model for some fixed dataset $\mathcal{D}$. Suppose function $f$ is $L$-Lipschitz, it can be shown that the variance of the sub-optimality performance gap of each model is bounded by the epidemic uncertainty:*

$$\mathrm{Var}\left(\Delta(p_{\theta_i}, y_k^*)\right) \leq c_2 L^2 d\sigma^2 + c_2 L^2 \Delta_{\mathrm{epistemic}}(y_k^*, \mathcal{D}), \ \ \forall i \in M, \tag{19}$$

*where $c_2$ is some universal positive constant.*

*Proof of Theorem 4.* At every iteration $k \in [K]$, let the target function value on which the conditional diffusion model conditions be $y_k^*$. The statement needs to hold for each conditional diffusion model in the ensemble, and thus for simplicity of notation, the subscript $i$ of $\theta_i$ is dropped in the remaining proof. With the existence of $\theta^* \sim p(\theta \mid \mathcal{D})$, we have $y_k^* = f(\boldsymbol{x}^*)$, where $\boldsymbol{x}^* \sim p_{\theta^*}(\cdot|y_k^*, \mathcal{D})$. Recall that $f(\boldsymbol{x}_k)$ is achieved by $\boldsymbol{x}_k \sim p_\theta(\cdot|y_k^*, \mathcal{D})$, where $\theta \sim p(\theta \mid \mathcal{D})$, and $N = 1$.

Thus, by Eve's law, the overall variance of $\Delta(p_\theta, y_k^*)$ can be decomposed as:

$$
\begin{aligned}
\mathrm{Var}\left(\Delta(p_\theta, y_k^*)\right) &= \mathrm{Var}\left(|y_k^* - f(\boldsymbol{x}_k)|\right) \\
&= \mathrm{Var}\left(|f(\boldsymbol{x}^*) - f(\boldsymbol{x}_k)|\right) \\
&= \underbrace{\mathbb{E}_{\theta,\theta^*}\left[\mathrm{Var}_{\boldsymbol{x}_k, \boldsymbol{x}^*}(|f(\boldsymbol{x}^*) - f(\boldsymbol{x}_k)| \mid \theta, \theta^*)\right]}_{T_1} + \underbrace{\mathrm{Var}_{\theta,\theta^*}(\mathbb{E}_{\boldsymbol{x}_k, \boldsymbol{x}^*}[|f(\boldsymbol{x}^*) - f(\boldsymbol{x}_k)| \mid \theta, \theta^*])}_{T_2}.
\end{aligned}
$$

In particular, the first term $T_1$ corresponds to the aleatoric component and the second term $T_2$ corresponds to the epidemic component. We then proceed to bound the above two terms separately.

**Step 1: bound $T_1$.** Under Assumption 2,

$$\mathrm{Var}_{\boldsymbol{x}_k, \boldsymbol{x}^*}(|f(\boldsymbol{x}^*) - f(\boldsymbol{x}_k)| \mid \theta, \theta^*) \leq L^2 \mathrm{Var}_{\boldsymbol{x}_k, \boldsymbol{x}^*}(\|\boldsymbol{x}^* - \boldsymbol{x}_k\| \mid \theta, \theta^*).$$

Under Assumption 3 and by Lemma D.2,

$$T_1 \leq L^2 \mathbb{E}_{\theta,\theta^*}[\mathrm{Var}_{\boldsymbol{x}_k, \boldsymbol{x}^*}(\|\boldsymbol{x}^* - \boldsymbol{x}_k\| \mid \theta, \theta^*)] \leq c_3 L^2 d\sigma^2. \tag{27}$$

**Step 2: bound $T_2$.** Under Assumption 2,

$$T_2 \leq L^2 \mathrm{Var}_{\theta,\theta^*}(\mathbb{E}_{\boldsymbol{x}_k, \boldsymbol{x}^*}[\|\boldsymbol{x}^* - \boldsymbol{x}_k\| \mid \theta, \theta^*]))$$

By Lemma D.1,

$$
\begin{aligned}
\mathrm{Var}_{\theta,\theta^*}(\mathbb{E}_{\boldsymbol{x}_k, \boldsymbol{x}^*}[|f(\boldsymbol{x}^*) - f(\boldsymbol{x}_k)| \mid \theta, \theta^*]) &\leq \mathrm{Var}_{\theta,\theta^*}\left(\mathbb{E}_{\theta,\theta^*}\left[8\sqrt{d}\sigma + \|\mathbb{E}_{\boldsymbol{x}^*}[\boldsymbol{x}^*] - \mathbb{E}_{\boldsymbol{x}_k}[\boldsymbol{x}_k]\|\right]\right) \\
&\leq \mathrm{Var}_{\theta,\theta^*}\left(\|\mathbb{E}_{\boldsymbol{x}^*}[\boldsymbol{x}^*] - \mathbb{E}_{\boldsymbol{x}_k}[\boldsymbol{x}_k]\|\right).
\end{aligned}
$$

Then by property of variance, we have

$$\mathrm{Var}_{\theta,\theta^*}\left(\|\mathbb{E}_{\boldsymbol{x}^*}[\boldsymbol{x}^*] - \mathbb{E}_{\boldsymbol{x}_k}[\boldsymbol{x}_k]\|\right) = \mathbb{E}_{\theta,\theta^*}\left[\|\mathbb{E}_{\boldsymbol{x}^*}[\boldsymbol{x}^*] - \mathbb{E}_{\boldsymbol{x}_k}[\boldsymbol{x}_k]\|^2\right] - \left(\mathbb{E}_{\theta,\theta^*}\left[\|\mathbb{E}_{\boldsymbol{x}^*}[\boldsymbol{x}^*] - \mathbb{E}_{\boldsymbol{x}_k}[\boldsymbol{x}_k]\|\right]\right)^2.$$

From the proof of Lemma D.2, we have

$$
\begin{aligned}
&\mathbb{E}_{\theta,\theta^*}\left[\|\mathbb{E}_{\boldsymbol{x}^*}[\boldsymbol{x}^*|\theta^*] - \mathbb{E}_{\boldsymbol{x}_k}[\boldsymbol{x}_k|\theta]\|^2\right] \\
&= \mathbb{E}_{\theta^*}[\mathbb{E}_{\boldsymbol{x}^*}[\|\boldsymbol{x}^*\|^2 \mid \theta^*]] + \mathbb{E}_\theta[\mathbb{E}_{\boldsymbol{x}_k}[\|\boldsymbol{x}_k\|^2 \mid \theta]] - 2\mathbb{E}_{\theta,\theta^*}[\mathbb{E}_{\boldsymbol{x}_k}[(\boldsymbol{x}_k|\theta)]^{\mathrm{T}}\mathbb{E}_{\boldsymbol{x}^*}[\boldsymbol{x}^*|\theta^*]] \\
&= 2(\mathbb{E}_\theta[\mathbb{E}_{\boldsymbol{x}_k}[\|\boldsymbol{x}_k\|^2 \mid \theta]] - \mathbb{E}_{\theta,\theta^*}[\mathbb{E}_{\boldsymbol{x}_k}[(\boldsymbol{x}_k|\theta)]^{\mathrm{T}}\mathbb{E}_{\boldsymbol{x}^*}[\boldsymbol{x}^*|\theta^*]])
\end{aligned}
$$

Combining the above results, we have

$$T_2 \leq L^2 \mathrm{Var}_{\theta,\theta^*}\left(\|\mathbb{E}_{\boldsymbol{x}^*}[\boldsymbol{x}^*] - \mathbb{E}_{\boldsymbol{x}_k}[\boldsymbol{x}_k]\|\right) \leq 2L^2 \mathrm{Var}_\theta(\mathbb{E}_{\boldsymbol{x}_k}[\boldsymbol{x}_k]). \tag{28}$$

Combining Equation (27) and Equation (28) completes the proof:

$$\mathrm{Var}\left(\Delta(p_\theta, y_k^*)\right) \leq c_3 L^2 d\sigma^2 + 2L^2 \mathrm{Var}_\theta(\mathbb{E}_{\boldsymbol{x}}[\|\boldsymbol{x}_k\|]).$$

$\square$

## D.2 Supporting Lemmas

**Lemma D.3.** *Let $\boldsymbol{x} \in \mathbb{R}^d$ be a $\sigma$-subGaussian ramdom vector, then*

$$\mathbb{E}[\|\boldsymbol{x} - \mathbb{E}[\boldsymbol{x}]\|] \leq 4\sigma\sqrt{d}. \tag{29}$$

**Lemma D.4.** *Let $\boldsymbol{x} \in \mathbb{R}^d$ be a $\sigma$-subGaussian ramdom vector, then its variance satisfies:*

$$Var[\boldsymbol{x}] \leq Cd\sigma^2, \tag{30}$$

*where C is some positive constant.*

*Proof of lemma D.4.* By definition of sub-Gaussian vector, for any direction $\boldsymbol{u} \in \mathbb{R}^d$ with $\|\boldsymbol{u}\| = 1$,

$$\mathbb{E}\left[\exp(\lambda\boldsymbol{u}^{\mathrm{T}}(\boldsymbol{x} - \mathbb{E}[\boldsymbol{x}]))\right] \leq \exp\left(\frac{\lambda^2\sigma^2}{2}\right), \quad \forall \lambda \in \mathbb{R}.$$

This implies that the second moment in any direction $\boldsymbol{u}$ satisfies:

$$\mathbb{E}\left[\boldsymbol{u}^{\mathrm{T}}((\boldsymbol{x} - \mathbb{E}[\boldsymbol{x}])(\boldsymbol{x} - \mathbb{E}[\boldsymbol{x}])^{\mathrm{T}})\right] \leq \sigma^2.$$

Therefore, the maximum eigenvalue of the covariance matrix is upper-bounded by $C\sigma^2$, where $C$ is some positive constant.

$$Var[\boldsymbol{x}] = \mathrm{tr}\left(\mathbb{E}\left[(\boldsymbol{x} - \mathbb{E}[\boldsymbol{x}])(\boldsymbol{x} - \mathbb{E}[\boldsymbol{x}])^{\mathrm{T}}\right]\right) \leq Cd\sigma^2.$$

$\square$

**Lemma D.5.** *In each iteration $k \in [K]$, let $\mathcal{D}$ be the collected dataset, $\theta$ and $\theta^*$ are parameters independently drawn from posterior $p(\theta|\mathcal{D})$, $\boldsymbol{x}_k \sim p_\theta(\cdot|y_k^*, \mathcal{D})$ and $\boldsymbol{x}^* \sim p_{\theta^*}(\cdot|y_k^*, \mathcal{D})$. For any measurable function $f$, and $\sigma(\mathcal{D})$-measurable random variable $\boldsymbol{x_k}$,*

$$\mathbb{E}\left[f(\boldsymbol{x}_k)\right] = \mathbb{E}\left[f(\boldsymbol{x}^*)\right].$$

*Proof of Lemma D.5.* Since the black-box function $f$ is measurable, and by the nature of Algorithm 1, in each iteration $k$, the generated sample $\boldsymbol{x_k}$, the target function value $y_k^*$, the predictive distribution $p_\theta(\cdot|y_k^*, \mathcal{D})$, the posterior distribution $p(\theta \mid \mathcal{D})$ are $\sigma(\mathcal{D})$-measurable at iteration $k$, the only randomness in $f(\boldsymbol{x})$ comes from the random sampling in the algorithm. Thus, condition on the training data $\mathcal{D}$ and target value $y_k^*$, by tower rule,

$$\mathbb{E}\left[f(\boldsymbol{x}_k)\right] = \mathbb{E}\left[\mathbb{E}\left[f(\boldsymbol{x}_k)|\theta\right]\right] = \int_\theta \int_{\boldsymbol{x}_k} f(\boldsymbol{x}_k)p_\theta(\boldsymbol{x}_k|y_k^*, \mathcal{D})p(\theta|\mathcal{D})\,\mathrm{d}\boldsymbol{x}_k\,\mathrm{d}\theta$$

$$= \int_\theta \int_{\boldsymbol{x}_k} f(\boldsymbol{x}_k)p_\theta(\boldsymbol{x}_k|y_k^*, \mathcal{D})\,\mathrm{d}\boldsymbol{x}_k\,p(\theta|\mathcal{D})\,\mathrm{d}\theta.$$

Note that both the true parameter $\theta^*$ and the chosen parameter $\theta$ are drawn from the same posterior distribution $p(\theta \mid \mathcal{D})$, we have

$$\int_\theta \int_{\boldsymbol{x}} f(\boldsymbol{x})p_\theta(\boldsymbol{x}|y_k^*, \mathcal{D})\,\mathrm{d}\boldsymbol{x}\,p(\theta|\mathcal{D})\,\mathrm{d}\theta = \int_{\theta^*} \int_{\boldsymbol{x}} f(\boldsymbol{x})p_{\theta^*}(\boldsymbol{x}|y_k^*, \mathcal{D})\,\mathrm{d}\boldsymbol{x}\,p(\theta^*|\mathcal{D})\,\mathrm{d}\theta^*.$$

As a result, we have

$$\mathbb{E}\left[f(\boldsymbol{x}_k)\right] = \int_{\theta^*} \int_{\boldsymbol{x}^*} f(\boldsymbol{x}^*)p_{\theta^*}(\boldsymbol{x}^*|y_k^*, \mathcal{D})\,\mathrm{d}\boldsymbol{x}^*p(\theta^*|\mathcal{D})\,\mathrm{d}\theta^* = \mathbb{E}\left[\mathbb{E}\left[f(\boldsymbol{x}^*)|\theta^*\right]\right] = \mathbb{E}\left[f(\boldsymbol{x}^*)\right].$$

$\square$

**Corollary 1.** *In each iteration $k \in [K]$, let $\mathcal{D}$ be the collected dataset, $\theta$ and $\theta^*$ are parameters independently drawn from posterior $p(\theta|\mathcal{D})$, $\boldsymbol{x}_k \sim p_\theta(\cdot|y_k^*, \mathcal{D})$ and $\boldsymbol{x}^* \sim p_{\theta^*}(\cdot|y_k^*, \mathcal{D})$. For any measurable function $f$, and $\sigma(\mathcal{D})$-measurable random variable $\boldsymbol{x_k}$,*

$$\mathbb{E}\left[\|\boldsymbol{x}_k\|\right] = \mathbb{E}\left[\|\boldsymbol{x}^*\|\right].$$

*Proof of Corollary 1.* Since the norm function is deterministic and $\sigma(\mathcal{D})$-measurable, the proof directly follows that of Lemma D.5. $\square$

# E  Optimality of Proposed Acquisition Function

**Theorem 1.** *Let $\mathcal{Y}$ be the constructed candidate set at each iteration $k \in [K]$ in Algorithm 1. By adopting UaE as the acquisition function to guide the sample generation process in conditional diffusion model, Diff-BBO (Algorithm 1) achieves a near-optimal solution for the online BBO problem defined in Equation (1):*

$$\max_{y_k \in \mathbb{R}} \sum_{k=1}^{K} f(\boldsymbol{x}_k), \ \ \boldsymbol{x}_k \sim p_\theta(\cdot \mid y_k, \mathcal{D}), \ \theta \in \Theta \ \Rightarrow \max_{y_k \in \mathcal{Y}} \sum_{k=1}^{K} \alpha(y_k, \mathcal{D}).$$

*Proof of Theorem 1.* Following Theorem 4, we can express the function evaluation as follows,

$$f(\boldsymbol{x}_k) = y_k - (y_k - f(\boldsymbol{x}_k)), \forall k \in [K].$$

The overall objective of the optimization problem defined in Equation (1) can then be further decomposed as

$$\max_{y_k \in \mathbb{R}} \sum_{k=1}^{K} f(\boldsymbol{x}_k), \ \ \boldsymbol{x}_k \sim p_\theta(\cdot \mid y_k), \ \theta \in \Theta$$

$$\Leftrightarrow \max_{y_k \in \mathbb{R}} \sum_{k=1}^{K} y_k - (y_k - f(\boldsymbol{x}_k)), \ \ \boldsymbol{x}_k \sim p_\theta(\cdot \mid y_k), \ \theta \in \Theta$$

$$\Rightarrow \max_{y_k \in \mathbb{R}} \sum_{k=1}^{K} y_k - \Delta(p_\theta, y_k).$$

By Theorem 4, which shows $\Delta(p_\theta, y_k^*)$ can be effectively upper bounded the epidemic uncertainty, we therefore have

$$\max_{y_k \in \mathbb{R}} \sum_{k=1}^{K} f(\boldsymbol{x}_k), \ \ \boldsymbol{x}_k \sim p_\theta(\cdot \mid y_k), \ \theta \in \Theta \Rightarrow \max_{y_k \in \mathcal{Y}} \sum_{k=1}^{K} y_k - \Delta_{\text{episdemic}}(y_k, \mathcal{D})$$

Essentially, our chosen acquisition function allows Diff-BBO to maximize the lower bound of the original optimization problem. $\qquad\square$

# F  Experiment Details

## F.1  Dataset Details.

**DesignBench (Trabucco et al., 2022)** is a benchmark for real-world black-box optimization tasks. For continuouse tasks, we use Superconductor, D'Kitty Morphology and Ant Morphology benchmarks. For discrete tasks, we utilize TFBind8 and TFBind10 benchmarks. We exclude Hopper due to the domain is known to be buggy, as explained in Appendix C in (Krishnamoorthy et al., 2023). We also exclude NAS due to the significant computational resource requirement. Additionally, we exclude the ChEMBL task because the oracle model exhibits non-trivial discrepancies when queried with the same design.

- **Superconductor (materials optimization).** This task involves searching for materials with high critical temperatures. The dataset comprises 17,014 vectors, each with 86 components that represent the number of atoms of each chemical element in the formula. The provided oracle function is a pre-trained random forest regression model.

- **D'Kitty Morphology (robot morphology optimization).** This task focuses on optimizing the parameters of a D'Kitty robot, including the size, orientation, and location of the limbs, to make it suitable for a specific navigation task. The dataset consists of 10,004 entries with a parameter dimension of 56. It utilizes MuJoCO (Todorov et al., 2012), a robot simulator, as the oracle function.

- **Ant Morphology (robot morphology optimization).** Similar to D'Kitty, this task aims to optimize the parameters of a quadruped robot to maximize its speed. It includes 10,004 data points with a parameter dimension of 60. It also uses MuJoCO as the oracle function.

- **TFBind8 (DNA sequence optimization).** This task seeks to identify the DNA sequence of length eight with the highest binding affinity to the transcription factor SIX6 REF R1. The design space comprises sequences of nucleotides represented as categorical variables. The dataset size is 32,898, with a dimension of 8. The ground truth is used as a direct oracle since the affinity for the entire design space is available.
- **TFBind10 (DNA sequence optimization).** Similar to TFBind8, this task aims to find the DNA sequence of length ten that exhibits the highest binding affinity with transcription factor SIX6 REF R1. The design space consists of all possible nucleotide sequences. The dataset size is 10,000, with a dimension of 10. The ground truth is used as a direct oracle since the affinity for the entire design space is available.

**Molecular Discovery.** A key problem in drug discovery is the optimization of a compound's activity against a biological target with therapeutic value. Similar to other papers (Eckmann et al., 2022; Jeon and Kim, 2020; Lee et al., 2023; Noh et al., 2022), we attempt to optimize the score from AutoDock4 (Morris et al., 2009), which is a physics-based estimator of binding affinity. The oracle is a feed-forward model as a surrogate to AutoDock4. The surrogate model is trained until convergence on 10,000 compounds randomly sampled from the latent space (using $\mathcal{N}(0, 1)$) and their computed objective values with AutoDock4. We construct our continuous design space by fixing a random protein embedding and randomly sampling 10,000 molecular embedding of dimension 32.

For each task, we arrange the offline dataset from Krishnamoorthy et al. (2023) in ascending order based on objective values and select data from the 25th to the 50th percentile as the initial training dataset. We prioritize data with lower objective scores to better observe performance differences across each baseline. The overview of all the task statistics is provided in Table 1.

| Task | Size | Dimensions | Task Max |
|---|---|---|---|
| TFBind8 | 32,898 | 8 | 1.0 |
| TFBind10 | 10,000 | 10 | 2.128 |
| D'Kitty | 10,004 | 56 | 340.0 |
| Ant | 10,004 | 60 | 590.0 |
| Superconductor | 17,014 | 86 | 185.0 |
| Molecular Discovery | 10,000 | 32 | 1.0 |

Table 1: Data Statistics

## F.2 Implementation Details.

We train our model on NVIDIA A100 GPU and report the average performance over 3 random runs along with standard deviation for each task. For discrete tasks, we follow the procedure in Krishnamoorthy et al. (2023) where we convert the $d$-dimensional vector to a $d \times c$ one hot vector regarding $c$ classes. We then approximate logits by interpolating between a uniform distribution and the one hot distribution using a mixing factor of $0.6$. We jointly train a conditional and unconditional model with the same model by randomly set the conditioning value to 0 with dropout probability of $0.15$.

For each task, we fix the learning rate at $0.001$ with batch size of 256. We use 5 ensemble models to estimate the uncertainty for our acquisition function. We set hidden dimensions to $1024$ and gamma to $2$. We use $10\%$ of the available data at each iteration as validation set during training.

## F.3 Ablation study

In this section, we conduct ablation studies to investigate the impact of our designed acquisition function, UaE. We compare Diff-BBO with the fixed condition approach. Instead of using UaE to dynamically determine which $y$ to condition on, the fixed condition approach always generates new samples conditioned on $w \cdot \phi_k$ (Line 9 of Algorithm 1) with a fixed weight $w$. As shown in Figure 4, Diff-BBO consistently outperforms the fixed condition approach. This demonstrates that our acquisition function is effective in identifying the optimal $y$ for conditioning.

Furthermore, we evaluate the effect of batch size, aka the number of queries per iteration on Diff-BBO on the Superconductor task. As shown in Figure 5, we compare the objective function score over number of function evaluations. We can see the performance of our approach remains similar when the batch size becomes small, suggesting remarkable robustness across different batch sizes. Hence,

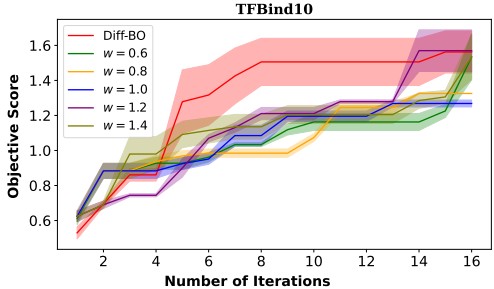
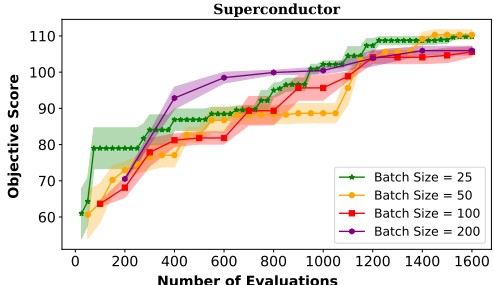

Figure 4: Impact of acquisition function design for black-box optimization on the TFBIND10 task. Comparison of Diff-BBO with five fixed-condition approaches, each with different conditioning weights. Results averaged across three random runs.

Figure 5: Ablation study to evaluate the effect of batch size on the superconductor task. The mean and standard deviation across three random seeds are plotted. Diff-BBO shows robust performances across different batch size given the same total number of evaluations.

Diff-BBO is a highly-scalable inverse modeling approach that can efficiently leverage parallelism to handle larger computational loads without compromising performance.

# G   Impact Statement

Optimization techniques can address various real-world problems, including drug and material design. Our method enhances sample-efficient online black-box optimization, potentially accelerating solutions in these areas. However, caution is needed to prevent misuse, such as optimizing drugs to enhance harmful side effects.

