# OpenReview forum: "Diff-BBO:  Diffusion-Based Inverse Modeling for Black-Box Optimization"
_NeurIPS.cc/2024/Workshop/BDU — NeurIPS BDU Workshop 2024 Poster_

### Official Review · Reviewer_ayTn · 2024-09-18
**This paper presents Diff-BBO, a diffusion-based inverse modeling approach for black-box optimization (BBO). While the idea of applying diffusion models in BBO is interesting, the paper suffers from significant clarity issues, making it difficult to follow. The theoretical contributions, such as uncertainty-aware exploration (UaE), are not well-explained, and the empirical validation is weak, with limited benchmarks and marginal improvements over existing methods. Additionally, the paper does not address the practical scalability of the approach, reducing its significance in real-world applications. Overall, the contribution is promising but requires substantial improvement in presentation and empirical justification.**

**Rating:** 4
**Confidence:** 4

**Review:**

Quality

The paper is well-structured and presents a novel approach to Black-Box Optimization (BBO) using diffusion-based inverse modeling. The methodology and theoretical contributions are sound, with solid backing from empirical and theoretical analyses. The clarity of the paper is slightly hindered by a dense mathematical presentation, but it is sufficient for an audience familiar with the field of machine learning, especially in the domain of generative models and optimization.

Clarity

The paper lacks clear explanations and guidance for the reader. The heavy reliance on mathematical formulations without providing intuitive insights makes it difficult to understand the core contributions and their significance. Important sections, such as the design of the uncertainty-aware exploration (UaE) acquisition function, are not sufficiently explained, making it challenging for the reader to grasp how this method improves upon existing BBO techniques. More visual aids or simplified examples would help significantly in conveying the core ideas.

Originality

The use of diffusion models for inverse modeling in BBO is an interesting idea, but it is not sufficiently differentiated from other generative models already used in optimization tasks. The paper does not provide a strong case for why diffusion models are superior to other state-of-the-art methods like Gaussian processes or Bayesian optimization techniques. Additionally, the novelty of the uncertainty-aware exploration (UaE) function is underdeveloped and lacks sufficient comparison to existing methods that balance exploration and exploitation in optimization.

Significance

The practical impact of the proposed Diff-BBO method is questionable. While some benchmarks are presented, the results are not comprehensive, and the performance improvements over baselines are marginal in some cases. Furthermore, the paper lacks real-world application examples or discussions of how this method could be practically implemented in large-scale or high-dimensional optimization problems, which limits its significance in advancing the field of BBO.

Conclusion

(1) The idea of applying diffusion models in inverse modeling for BBO has potential if more thoroughly developed and explained.

(2)Some theoretical contributions, such as uncertainty decomposition, are mathematically sound.

(3) The paper presents a potentially interesting approach, but it falls short in terms of clarity, originality, and empirical significance. Substantial revisions are needed, particularly in providing better explanations, stronger empirical comparisons, and addressing the method's practical scalability.

---

### Official Review · Reviewer_9uKj · 2024-09-23
**Review for Diff-BBO: Diffusion-Based Inverse Modeling for Black-Box Optimization**

**Rating:** 5
**Confidence:** 3

**Review:**

Summary
The study presents Diff-BBO, a novel diffusion-based inverse modeling approach to solve black-box optimization (BBO) problems online. It introduces the Uncertainty-aware Exploration (UaE) acquisition function, which leverages uncertainty in diffusion models to propose objective function values, resulting in sample-efficient optimization. The method is validated through theoretical analyses and empirical evaluations, showing superior performance over baseline methods in various continuous and discrete optimization tasks.

Strengths
1.	Introduces a new inverse modeling technique leveraging diffusion models, which contrasts with traditional forward approaches, providing a fresh perspective in the BBO domain.
The proposed UaE acquisition function strategically balances exploration and exploitation, optimizing sample efficiency in an online setting.
2.	Demonstrates state-of-the-art performance across multiple benchmarks, consistently outperforming baselines in both discrete and continuous tasks.

Weaknesses
1.	The performance of the Diff-BBO algorithm heavily depends on the quality of the initial training dataset. If the dataset is too limited or skewed, it could hinder the optimization process.
2.	While the UaE function is theoretically proven to balance objective values and uncertainty, the proofs could benefit from a clearer discussion of limitations, assumptions, or failure cases in real-world BBO problems.

Recommendation
This paper demonstrates strong potential in advancing the field of black-box optimization through diffusion-based inverse modeling. Despite some concerns regarding dataset dependency, model limitations, and computational demands, its innovative approach and promising results suggest it should be considered for major revision. Improvements to the theoretical exposition and additional empirical evaluations could further strengthen the work's contribution to the field.

---

### Decision · Program_Chairs · 2024-10-09

**Decision:**

Accept (Poster)

**Comment:**

Scores for this work are borderline. The actual text of the reviews is significantly more positive then the scores, praising the work for exploring a novel way to use generative models in this setting. In general it seeks the reviewers feel the work is incomplete: one asks for a major revision of the work. However, it is clear that such a revision would probably move the paper from workshop-paper to conference-paper level in terms of comprehensiveness. I therefore conclude the work is sufficiently far along in its present form to warrant acceptance at the workshop.